# Explaining Bounding Boxes in Deep Object Detectors Using Post Hoc Methods for Autonomous Driving Systems

**DOI:** 10.3390/s24020516

**Published:** 2024-01-14

**Authors:** Caio Nogueira, Luís Fernandes, João N. D. Fernandes, Jaime S. Cardoso

**Affiliations:** 1Faculty of Engineering, University of Porto, 4200-465 Porto, Portugal; up202103249@edu.fe.up.pt (L.F.); jdfernandes@fe.up.pt (J.N.D.F.); jsc@fe.up.pt (J.S.C.); 2Instituto de Engenharia de Sistemas e Computadores, Tecnologia e Ciência, 4200-465 Porto, Portugal

**Keywords:** explainable AI, autonomous driving, object detection

## Abstract

Deep learning has rapidly increased in popularity, leading to the development of perception solutions for autonomous driving. The latter field leverages techniques developed for computer vision in other domains for accomplishing perception tasks such as object detection. However, the black-box nature of deep neural models and the complexity of the autonomous driving context motivates the study of explainability in these models that perform perception tasks. Moreover, this work explores explainable AI techniques for the object detection task in the context of autonomous driving. An extensive and detailed comparison is carried out between gradient-based and perturbation-based methods (e.g., D-RISE). Moreover, several experimental setups are used with different backbone architectures and different datasets to observe the influence of these aspects in the explanations. All the techniques explored consist of saliency methods, making their interpretation and evaluation primarily visual. Nevertheless, numerical assessment methods are also used. Overall, D-RISE and guided backpropagation obtain more localized explanations. However, D-RISE highlights more meaningful regions, providing more human-understandable explanations. To the best of our knowledge, this is the first approach to obtaining explanations focusing on the regression of the bounding box coordinates.

## 1. Introduction

Object detection is a computer vision task for several domains to localize and identify objects within an image. In the autonomous driving (AD) context, object detection consists of a perception task that aids the system in understanding its surrounding environment. The architectures of neural models that perform this task are dense [1,2], with many convolutions that extract features from the input image and subsequent operations that identify the objects regarding their localization and classification. Consequently, the networks make decisions in a black-box manner.

The autonomous driving context forces deep learning models to make critical decisions affecting the safety of their users, which makes it essential to consider legal aspects [3]. As such, the need to obtain explanations for the inferences performed by these models emerges. Furthermore, researchers can better grasp the model’s behavior, especially when it returns wrong predictions.

Explainable AI (xAI) is a research topic that tackles the challenge of developing models whose decisions can be understood by humans. In the context of autonomous driving, these techniques increase trustworthiness, transparency, and accountability for the state-of-the-art black box models. Its relevance has been increasing abruptly across several domains, including computer-vision-related tasks or other general deep learning models (e.g., health [4], vehicle networking [5]).

The body of knowledge concerning xAI techniques applied to driving contexts is mainly focused on the decision-making process of the system, which results from all operations performed by these systems (e.g., perception, localization, planning, control, and management). Moreover, the data that flows in the models used for autonomous driving tasks also need to be exchanged with some infrastructure that serves as a base station [6]. Furthermore, xAI techniques for the perception part of these systems could be more extensive. Moreover, most techniques target class-discriminative explanations (i.e., identifying the most critical aspects to classify an object with a particular class). This work aims to take existing perception-based explanation techniques as a starting point and adapt them to the bounding box coordinates regression context. This first approach will explore visual explanations through model-agnostic methods, namely gradient and perturbation-based techniques. Ultimately, the goal consists of obtaining meaningful interpretations that convey information about the network’s reasoning behind the localization of the detections. The approaches developed with this goal in mind will improve state-of-the-art explainability methods to produce more comprehensive information regarding the output of the model being explained. In the context of autonomous driving, there are no scientific contributions that leverage the techniques used in this article and adapted from other domains to produce explanations for the regression part of the object detection task, i.e., the prediction of the localization of the bounding box. As such, this paper will extend what we have seen previously regarding explanations for perception models. In addition, the techniques explored here can be readily applied to several models. On the other hand, this work can also be extended to provide even richer interpretations (e.g., leveraging multi-modal data).

In addition, the explanations produced should be evaluated and compared using subjective observations and performance metrics. This assessment phase helps to identify the aspects that play an essential role in the quality of the explanations. As such, the experimental setup should have varying characteristics concerning the object detectors explained, datasets used, and hyperparameters of each method.

This article is structured into five sections. The introduction (Section 1) gives a slight overview of the autonomous driving context and the xAI field. Section 2 explores different techniques in the literature that aim to achieve explainability in computer-vision tasks. Section 3 explains the approaches used to tackle the challenge at hand, including the theoretical fundamentals behind each strategy. Section 4 shows the results obtained for each technique, establishing a comparison that takes into consideration different aspects important to assess the quality of explanations generated. Finally, Section 5 concludes the study, referring to its findings and future directions that the work may follow.

## 2. Related Work

Before jumping into explainable AI methods, it is important to fully understand the challenges inherent to object detection. This computer-vision task is widely studied in the context of real-world problems across several domains (e.g., medicine, autonomous driving, agriculture, retail). Object detection models must be able to identify and localize the objects within an image, using point cloud or RGB data. As such, instead of using a single loss function that guides the backpropagation during training, object detection needs at least two different losses, i.e., classification loss and localization loss.

Object detection architectures can be categorized into two types. The one-stage-detector family is led by YOLO [2], and is able to locate and classify objects simultaneously. Therefore, these models are often time-efficient, outperforming their counterparts. On the other hand, the two-stage-detectors consist of a Region Proposal Network (RPN) that extracts Regions of Interest (ROIs), which are then used to classify the object contained within it. The backbone is the initial feature extraction step that exists in all architectures. This module of the network is responsible for learning and obtaining useful features that the subsequent layers will use to make decisions. Autonomous driving models leverage object detection models to understand the environment in which the vehicle is placed. Thus, these models are crucial to the perception phase of autonomous driving and will be thoroughly studied in this paper.

Regarding explainable AI (xAI) targeting autonomous driving, the literature studies all the different stages of an autonomous driving model. Reference [7] presents an extensive review of several techniques that can be applied to deep-neural-network-based models included in the perception, decision-making, and even in the information security of autonomous vehicles. However, the literature regarding the explaining of perception models, and more specifically, of the regression of bounding boxes, could be extended.

In the xAI literature, there is a difference between explaining a neural network and developing an intrinsically explainable one. Chen et al. [8] developed ProtoPNet, which obtains global explanations for image classification tasks in an inherently explainable manner by leveraging information in prototypes learned during optimization. These prototypes consist of a representation of each class in the latent space. The ProtoPNet method established itself as one of the main techniques for obtaining inherently explainable models. Several subsequent contributions consist of slight improvements to the original formula (e.g., [9,10,11]).

On the other hand, techniques known as model agnostic techniques consist of methods applied after the model has been adequately trained. These methods can be applied to any model, regardless of the specific algorithm or architecture used, and typically involve analyzing the input–output relationship of the model. The main benefit of being independent of the model is preserving its original performance.

Gradient-based methods perform the retro-propagation of information in the network, evaluating the gradient of the output to compute a heat map consisting of the most contributing regions. Zeiler and Fergus [12] proposed a visualization technique called deconvnet, that reproduces individual feature maps at any network layer. More recently, deep learning researchers developed improved versions of this technique [13]. Moreover, the study by Das and Rad [14] contains an extensive review of xAI techniques applied to several domains, including autonomous driving. The explanations explored in the latter survey are class-discriminative.

Grad-CAM [15] is one of the methods used for obtaining visual explanations. This approach leverages information contained in the gradient of the output, producing a heatmap that highlights the most important features of the outcome returned by the model. The gradients consider the predicted score for class *c* as the output ($y^{c}$). As such, the calculation is performed with this score with respect to the feature map activation at a particular convolutional layer ($A^{k}$). The weights are then global average pooled and normalized, resulting in a single-dimension vector with length *K*, equivalent to the number of channels in the feature map returned by the considered convolutional layer. The latter vector consists of *K* weights, that will multiply by the *K* feature maps obtained through the normal process of forward-propagation across the backbone. The final heatmap results from a weighted addition between the feature maps multiplied by the weights. Guided backpropagation [16] has gained considerable attention for its ability to highlight the important features and regions within an input that influence the network’s decision-making process. Similar to Grad-CAM, this approach takes as a parameter a value for which the influence of the pixels in the input image is observed. In the related literature, tackling the usual problems of Grad-CAM with Smooth Grad [17] is common. This technique introduces a form of regularization by leveraging the concept of gradient noise to enhance the quality and reliability of the model’s predictions.

Perturbation-based methods slightly modify the features in a specific region of the input and observe the behavior of the network in these new circumstances. Petsiuk et al. [18] proposed the RISE method, which uses random masks to compute similarity scores that produce a final heatmap. Petsiuk et al. [19] consists of a subsequent iteration of this method that allows more information to flow into the explanation. These perturbation-based approaches have not been applied to the autonomous driving context.

Local approximation methods approximate the behavior of a machine learning model around a specific input data point by fitting a simpler model to the model’s output in the vicinity of that point. In Local Interpretable Model-agnostic Explanations (LIME) [20], the simple model uses a weighting scheme based on the distance of each data point from the point of interest. SHAP (SHapley Additive exPlanations) [21] uses a game-theoretic approach based on the Shapley values from cooperative game theory.

In addition, counterfactual analysis approaches can also be employed to understand the decision-making reasoning of the neural network. In counterfactual analysis, researchers seek $x^{\prime }$, a modified version of *x*. With this new input, it is possible to define which input features contributed to the outcome of the classification. There have been some contributions related to this technique [22], aiming to reduce both the distance between predictions and the distance between the inputs.

## 3. Proposed Framework

The present section explores the different experimental setups used in this work. Section 3.1 performs an in-depth analysis of the datasets chosen, as well as the detectors, which will then be explained in the later stages. The remaining subsections propose different techniques, which are adapted from other domains to better fit the context of autonomous driving.

### 3.1. Training Procedure

The models are a crucial part of the experimental setup. Therefore, they must be appropriately trained and evaluated before moving onto the explanation part of the work, as it focuses solely on model-agnostic techniques. The dataset selection was carefully tailored to suit the requirements of the project, with a deliberate choice to utilize the KITTI autonomous driving benchmark [23]. Additionally, the Pascal VOC 2012 [24] dataset was employed for validation purposes, given its widespread use and recognition in the relevant academic literature. Table 1 contains exploratory data analysis regarding the characteristics of each dataset. As we can see, the autonomous driving context is significantly distinct from other datasets studied in the object detection literature. From a general perspective, KITTI images contain more objects (primarily cars) with a lower image area than Pascal VOC images. The characterization of each dataset is vital in the study carried out, as the quality of explanations can be volatile according to the input image used in the inference. Regarding the training setup, both datasets are divided into three sets, with the train set representing 80% of the data, while both the validation and the test set are assigned 10% of the dataset being used.

This work employs the Faster-RCNN two-stage detector [1]. For comparison purposes, several models were trained against each dataset. Moreover, different backbones were used, varying the number of convolutional layers. Several Faster-RCNNs were trained using the ResNet [25] architecture. The training process was performed for each dataset using ResNets with 50 and 101 convolutional layers. Additionally, a separate model was trained, considering solely the regression losses for the optimization. The goal of including this particular network in this study is to observe whether this regression-focused optimization brings advantages to the explanations obtained, especially in gradient-based methods that directly leverage features captured in the backbone.

Table 2 shows the test performance of all models with respect to the mean Average Precision (with 50% IoU threshold) and the smooth L1 loss. These metrics are also considered during the training of the Faster-RCNN, hence their inclusion in this comparison. The results obtained are aligned with state-of-the-art studies using Faster-RCNN for both datasets. As can be seen, the regression-focused model achieves the best performance concerning the smooth L1 loss. The difference between the mAP values is related to the number of classes between datasets, i.e., Pascal VOC contains 19 classes, while KITTI contains 9. The training setup was successful because all detectors achieved satisfactory results, which indicates that the models are learning meaningful patterns. This ultimately guarantees that the explanation techniques, which are the focus of this work, will not be limited by the models’ behavior.

The xAI methods studied in this paper are versatile in the sense that they can be readily used with different object detection architectures. In fact, this is an advantage of using model-agnostic xAI. Consequently, for comparison reasons, it is also crucial to include in this study a one-stage detector. As such, a YOLO model was trained against the KITTI dataset. The training setup for the latter model was different from all the ones listed in Table 2, as no early stopping was employed, and the train ended after 20 epochs, which is roughly the same number of iterations that the Faster-RCNN-based detectors took to converge. Moreover, the train, validation, and test split had the same proportions, but the sets were different. Regarding the performance achieved by YOLO, the mAP was 0.647. The regression loss used by YOLO is the Mean Squared Error (MSE) loss, instead of the Smooth L1 loss used by Faster-RCNN. In our setup, the YOLO model obtained an MSE loss of 1.168.

Despite the fact that YOLO is a modern architecture, the results with respect to normal object detection and machine learning metrics are worse for the one-stage detector. In this setup, this might be related to the significant number of small objects in the images, i.e., cars are often far from the camera. This is a known issue of YOLO, as the division of the scene into a grid can introduce difficulty in identifying objects that do only appear in a small portion of the image.

### 3.2. Grad-CAM

Grad-CAM is a gradient-based technique for obtaining visual explanations, as explored in Section 2. Its versatility makes it easier to implement in any object detector and makes it suitable for multiple contexts. In this context, we use Grad-CAM as a first approach to generate saliency maps. The remaining gradient-based techniques derive some of the Grad-CAM’s mechanisms.

The overview of the experimental setup described is shown in Figure 1. The image flows in an unidirectional way since it is picked from the validation set, passing through an inference achieved by the detector, and finally, using the resulting feature maps to compute the final heatmap.

Bounding boxes are rectangles identified by two pairs of coordinates. As such, applying the Grad-CAM method to highlight the features important in localization requires adaptations to the original algorithm. The original algorithm relies on class scores $y^{c}$ to produce heatmaps, resulting in class-discriminative saliency maps. As such, we propose applying Grad-CAM using two scalar transformations as the output of the gradients, which are computed with respect to the parameters of the last convolutional layers. The aforementioned transformations consist of the euclidean diagonal distance and the geometric slope given by the coordinates that identify the bounding box. Thus, the heatmap produced by the Grad-CAM method identifies features that are positively correlated to both measures for a particular detection in the input image. Throughout the paper, whenever a transformation T(bbox) is mentioned, that means one of the methods transforms the bounding box into a scalar, i.e., either the euclidean distance or the slope.
(1)$$W_{k}=\frac{1}{Z}\sum_{i}\sum_{j}\frac{\partial T(bbox)}{\partial A_{i,j}^{k}}$$

Equation (Equation 1) formalizes our approach to Grad-CAM. T(bbox) symbolizes the transformations made to convert the bounding box into a scalar value; $A^{k}$ represents the feature map of the *k*th convolutional layer; and *Z* is the normalization factor applied to the weights. Mathematically, the saliency maps obtained with the weights resulting from the latter formula convey information concerning the physical attributes of the explained bounding box.

### 3.3. Guided Backpropagation

Guided backpropagation is another prominent gradient-based technique that involves computing gradients and leveraging them to produce an interpretation heatmap, as explained in Section 2.

The difference between this approach and Grad-CAM is in the computation of the gradients. As explored in Section 3.2, grad-CAM computes the gradients of the outputs with respect to the weights of a particular convolutional layer in the backbone, while gradient backpropagation projects these gradients to the input image. Consequently, localization issues resulting from the final upsampling step in grad-CAM do not interfere with the results in guided backpropagation.

Our approach to guided backpropagation uses the transformation methods discussed in Section 3.2 to compute the gradients, resulting in heatmaps that show the important areas for the localization of the bounding boxes.

### 3.4. Smooth Grad

Smooth Grad is a Grad-CAM-derived model that obtains several interpretations using the latter method and smoothens the noise that is captured. The context of autonomous driving is intrinsically noisy, with many objects interfering with the quality of the explanations generated. Consequently, considering the nature of Smooth Grad and the problems that it intends to solve, it is important to check whether it is a viable alternative to Grad-CAM despite needing heavier computational effort due to the need for multiple backpropagations.

The algorithm applied takes the instructions given in the original paper and follows the majority of them while making some modifications to better fit the context of this work. The noise level (σ) is the standard deviation used to generate random floating values that should be added to the original image. The authors of the original paper conclude that a value between 10% and 20% is optimal for maintaining the image’s structural aspects and sharpening the generated heat maps. In this work, the σ value was set to 15%. Regarding the sample size (*n*), the authors also observed little to no smoothening for cases where more than 50 perturbed images are analyzed. Despite setting *n* to 50 in an initial phase, the empirical results point out that, in this autonomous driving context, the sample size can be lowered even further without significant differences regarding the results’ quality. As such, 25 is a comfortable sample size that balances the computational effort required per image and the quality of explanations generated.

### 3.5. Contrastive Explanations

The application of Grad-CAM using simple numerical transformations to the bounding box structures often produces vague explanations. Thus, the idea of employing visual counterfactual explanations to generate heatmaps more aligned with human explanations emerges. This approach requires the existence of a contrastive detection that ultimately aids the contextualization of the regions activated in the final heatmap. In this work, contrastive bounding boxes are produced via simple translations and scaling to the original bounding box predicted by the model. Figure 2 illustrates the production of a contrastive example from a particular bounding box to be explained. Our method strives for counterfactual examples that consist of sliding the original prediction so that the manufactured example is mislocalized by a small but noticeable margin. In addition, scaling can also be applied so that the contrastive bounding box can be bigger or smaller than the original prediction.

Prabhushankar et al. [26] introduce the idea of obtaining contrastive explanations with grad-CAM. In the paper, the authors calculate the gradients of relative loss between two classes with respect to the model’s parameters. In this work, we propose adapting this technique to the regression context by using the smooth L1 loss instead of the cross-entropy loss originally used for the class-discriminative explanations obtained in the aforementioned paper.

Figure 3 summarizes the Grad-CAM approach using contrastive explanations. The method is similar to that explained in Section 3.2, with the main difference being the formula used to obtain the weights. The existence of a second bounding box removes the need for transforming the predicted rectangle into a scalar value, as the difference between both detections is given by a relative loss function that is already well established in the object detection literature.

### 3.6. Perturbation-Based Approach

Perturbation-based methods attempt to obtain explanations by iteratively generating noise to the input image and observing the model’s behavior in these new circumstances. Petsiuk et al. [18] created RISE (Randomized Input Sampling for Explanation), which systematically eliminates from the heatmap regions that are irrelevant to the detection. Petsiuk et al. [19] is a subsequent technique based on RISE that generates explanations similarly. However, the detection vectors computed after each inference determine the information that flows into the saliency map. The authors use detection vectors comprising three elements for each detection, comprising localization and classification information, and an additional objectness score (Equation (Equation 2)). In our work, the similarity scores’ computation method is the same as the one proposed by the authors of the D-RISE paper.
(2)$$d_{i}=\left[L_{i},P_{i},O_{i}\right]=\left[\left(x_{min},y_{min},x_{max},y_{max}\right),\left(p_{1},\cdots ,p_{c}\right),O_{i}\right]$$

The images are perturbed with randomly generated masks that obscure some of their original regions. The saliency maps result from inferences made on the masked images, as the model returns values that are used to compute a detection vector. The weights of this operation are the similarity scores between the detection vector of the bounding box predicted on the original image and that of its corresponding detection on the image perturbed by the mask. Furthermore, this method was tested on the MS COCO dataset [27], which presents characteristics similar to Pascal VOC and offers an environment different from the autonomous driving context. As such, a modification was made to the hyperparameters used by the D-RISE method, namely regarding the mask generation process: the probability threshold (*p*) was reduced from 0.5 to 0.25 to prevent the occurrence of masks that entirely occlude object detection. Before this adjustment, it was empirically verified that several iterations were irrelevant, especially when explaining detections with a lower area.

In this work, two separate experimental setups were used: one of them considered the classification information while the other did not. This approach establishes a comparison regarding the influence of classification similarity on the localization capability of the saliency maps generated.

Furthermore, D-RISE takes advantage of including detection vectors to study detector failure cases. The authors compute the difference between saliency maps to outline regions wrongly considered relevant by the model. In this work, we leverage this technique to generate contrastive explanations. Instead of computing a saliency map with respect to the ground-truth bounding box, we generate a separate heatmap for a counterfactual example using the technique explained in Section 3.2. The contrastive explanation is obtained with the difference between normalized activation maps related to the predicted bounding box and the contrastive example.

## 4. Results and Discussion

The work carried out focuses solely on visual explanations. Consequently, the analysis and evaluation regarding the quality of the results produced should be primarily visual and subjective (Section 4.1). However, to address possible bias in the assessment of explanations, a numerical metric is introduced and applied in the results in Section 4.2.

### 4.1. Visual Evaluation

Regarding visual evaluation, due to the context of this work, most examples should evaluate the ability of methods to provide relevant and understandable explanations in the autonomous driving context.

Figure 4 shows an overview of the results obtained using the Grad-CAM setup on the autonomous driving dataset. Overall, Grad-CAM is known for its noise vulnerability and spacial sensitivity. Moreover, increasing the number of convolutional layers enhances noise propagation throughout the network. The high number of calculations during forward propagation makes it hard to project the features back into the original input dimension. These limitations of Grad-CAM and gradient-based methods generally cause some explanations to have characteristics similar to those observed in Figure 4d–f.

Figure 5 contains the results obtained with the guided backpropagation method for an input image. The saliency map shown in the latter figure was obtained with the distance transformation and the regression-focused model, as this combination achieved better results according to an initial subjective visual assessment. Guided backpropagation heatmaps are dispersed, activating a low number of pixels around the region where the object is located. This technique does not perform any upsampling, as no feature maps are used, as opposed to Grad-CAM. Consequently, localization problems introduced by linear interpolation are not considered. Despite successfully identifying pixels near the object, explanations are not intuitive or human-understandable, as they highlight pixels instead of proper regions that positively influence the detection.

Figure 6 contains the results obtained through SmoothGrad and establishes a comparison to the grad-CAM method. The visual observation concludes that this iterative process reduces the activated pixels. Figure 7a shows a localized explanation, where grad-CAM does not capture noise introduced by other objects in the image. The smoothGrad algorithm applied to this context reduces the number of activated pixels (Figure 7b). On the other hand, the noise reduction capabilities of this algorithm are tested in the second image studied, where the grad-CAM explanation contains noise in the region that belongs to an irrelevant object (Figure 7c).

On the other hand, empirical observations concluded that the SmoothGrad technique can only succeed when the original grad-CAM heatmap achieves some localization capabilities.

Figure 7 outlines the empirical results obtained with the contrastive grad-CAM technique for both datasets, using different backbone architectures. The regions highlighted in red represent features positively related to the increase of the relative loss between both detections (i.e., areas that help the model decide for the predicted bounding box instead of the contrastive example). On the other hand, blue zones represent a negative (or null) correlation between the corresponding features and the relative loss. In addition, in the autonomous driving context (Figure 7c,d), the noise level interfering with the saliency map production is seemingly superior. However, even in the latter case, which is naturally more demanding to explain, the method identifies regions within the predicted bounding box as not positively correlated to the loss increase. On the other hand, the activations are gradually increasing in the outer areas of the objects explained.

Figure 8 shows example explanations generated via the D-RISE method. Both experimental setups obtain saliency maps with satisfactory localization capabilities (i.e., most activated regions are located inside or close to the bounding box explained). Moreover, the D-RISE method effectively reduces the noise captured in the explanations, as regions far from the detection have close to null activation values.

Figure 9 contains a contrastive explanation produced with this approach. The interpretation of this heatmap is different from the one explained for gradient-based methods; most activated pixels should be located in regions that exclusively belong to the predicted detection. D-RISE heatmaps are weighted with similarity scores, whereas gradient-based techniques use the loss as the output for the gradients that act as weights for the saliency maps.

Both families of xAI methods explored in this subsection focus on a single object per saliency map, which can be a signnificant issue given the nature of autonomous driving. This limitation can be overcome by producing multiple visual explanations for each analyzed scene. With this approach, it becomes possible to explain inferences made on multiple objects with the objects being explained separately.

### 4.2. Numerical Assessment

The visual explanations obtained from techniques explored in Section 3 produce visual explanations and, as such, demand a primarily visual evaluation. However, this type of subjective assessment can introduce bias in the evaluation. To address this problem, an Explainable AI metric must be employed and introduced in this context to aid in evaluating the saliency maps. In this work, the metric chosen was the Pointing Game measure [28], which explains the whole validation set and verifies in each heatmap whether the most activated pixel is contained within the bounding box (hit). The final value given by the metric consists of the proportion of success cases (hits) in the total amount of samples tested (number of images in the validation set). The choice of this particular metric makes sense in the context of this work, as all explanations generated consist of activation maps. Moreover, it solely takes into consideration the localization aspects of the evaluation. In summary, the Pointing Game metric is capable of evaluating the consistency of the xAI methods to provide localized explanations across a significant number of test cases, e.g., the whole validation set.

Table 3 compares all the experimental setups used for Grad-CAM explanations. The distance transformation achieves better results than the slope operation, as it is generally more effective in describing the characteristics of a particular detection. As expected, the regression-focused model obtains more localized explanations as it probably benefits from the patterns learned during optimization that minimized the regression losses.

Table 4 depicts the results from gradient and perturbation-based methods, comparing these techniques. According to the results, gradient-backpropagation achieves better results concerning the Pointing Game metric, with D-RISE obtaining a close performance. On the other hand, it is verified that using detection vectors that solely encode localization information is beneficial in this context. Lastly, the latter table concludes that contrastive explanations are much more localized when produced via the perturbation-based approach.

The numerical results shown in the two tables mentioned above should not serve as a stand-alone evaluation regarding the quality of explanations. Despite guided-backpropagation being the method with better numerical results, its visual quality is minimal since the saliency maps are localized but still highly dispersed. On the other hand, D-RISE produces results with similar localization capability and highlights regions in a much more human-understandable manner, with little to no noise being captured. On the other hand, the high number of iterations needed for each explanation makes the perturbation-based approach much slower than gradient-based methods, i.e., each iteration performs the forward propagation of the perturbed image into the network. In contrast, gradient-based approaches need only a single propagation through the detector.

Furthermore, Table 4 also shows results regarding the YOLO architecture. The experimental setup for this step solely considered localization information for the D-RISE detection vectors, as this method achieved the best overall performance in the Faster-RCNN-based models. However, the saliency maps gathered for obtaining the latter result were computed with respect to a validation set that was different from the one used by detectors in the same table. Moreover, the nature of Pointing Game itself makes it a good metric to establish this comparison, even across different architectures. Even though the comparison was made in different setups, we still consider it a trustworthy manner of effectively comparing the D-RISE computed with YOLO in the global scope of the work carried out.

There is a significant difference between the results obtained for the YOLO model and the remaining two-stage detectors. This disparity does not reflect different behavior in the D-RISE algorithm for the models. Rather, it mirrors the discrepancy in the inferring capacity of these models (Section 3.1). The nature of post hoc perturbation-based methods make them maintain similar behaviors when exposed to different setups, i.e, the visual explanations in Section 4.3 are similar to those in Section 4.1.

### 4.3. Other Architectures

As stated in Section 3.1, a YOLO model was trained, so that its results could be explained with the xAI methods explored. As visual explanations obtained through D-RISE were significantly better than those produced by gradient-based explanations, YOLO results were explained using the perturbation-based approach.

Figure 10 shows the results of D-RISE when operating against inferences performed by the one-stage detector trained in our experimental setup. The visual quality of the saliency map is seemingly similar to those produced using two-stage detectors, which is quite satisfying. Moreover, it is also important to highlight that D-RISE can successfully produce explanations regarding different objects within the same image, even if they are close to one another. Such results could not be depicted by gradient-based techniques, as the presence of objects close to the detection being studied is sufficient to introduce noise to the saliency map, which will ultimately result in visual explanations that are not understandable.

## 5. Conclusions

The study extensively experimented with different explanation techniques and distinct setups for each approach. Most of the algorithms used consist of previously existing methods adapted to the regression context. In gradient-based methods, this adaptation meant computing scalar transformations from the bounding box object. In contrast, in D-RISE, the algorithm was slightly changed so that explanations better reflect the most important regions for the localization of each object. Moreover, contrastive explanations allowed a different way of explaining object detection that was more intuitive and human-understandable.

The main disadvantage of the D-RISE approach is the computational effort needed to generate a single explanation. However, the visual explanations produced with this perturbation-based approach are consistent and provide human-understandable interpretations in the vast majority of cases. As such, this approach is better than the much faster alternatives provided by gradient-based explanations, especially if the system is not concerned with the overhead introduced by the xAI module.

The comprehensiveness of the work carried out allowed a meaningful comparison between all methods used. It ultimately fulfilled the original goal of establishing an initial study regarding xAI in object detection for the context of autonomous driving. We can draw from this study that the challenges offered by the autonomous driving context require complex explanation methods. The lower area and higher number of objects studied significantly impact the saliency maps produced. As such, especially in applications where time is not a constraint, D-RISE offers better explanations even with more challenging images.

## Figures and Tables

**Figure 1 sensors-24-00516-f001:**
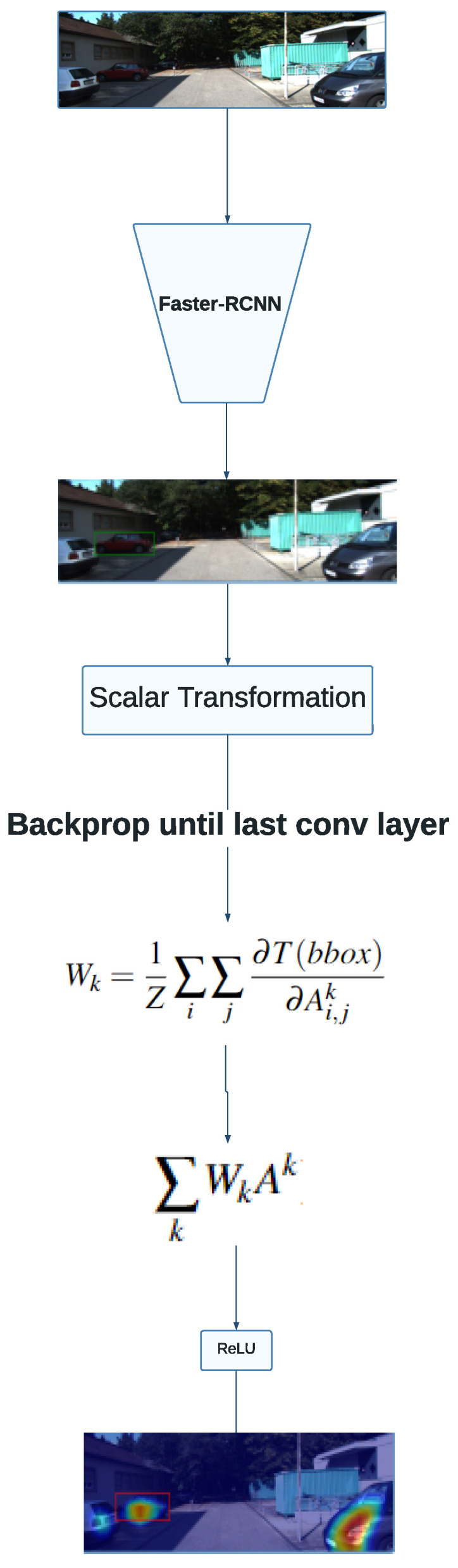
Diagram of the experimental setup used in this section. Red regions contain higher activation values, whereas blue regions represent regions with lower activation values.

**Figure 2 sensors-24-00516-f002:**
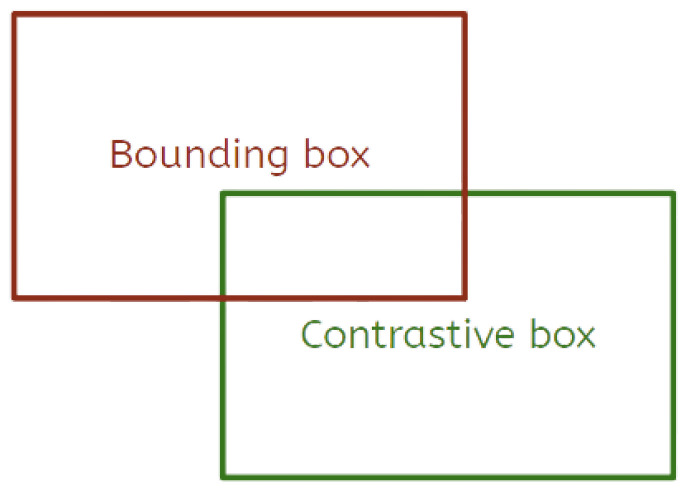
Illustration of the method used to produce the contrastive bounding box starting from a predicted detection.

**Figure 3 sensors-24-00516-f003:**
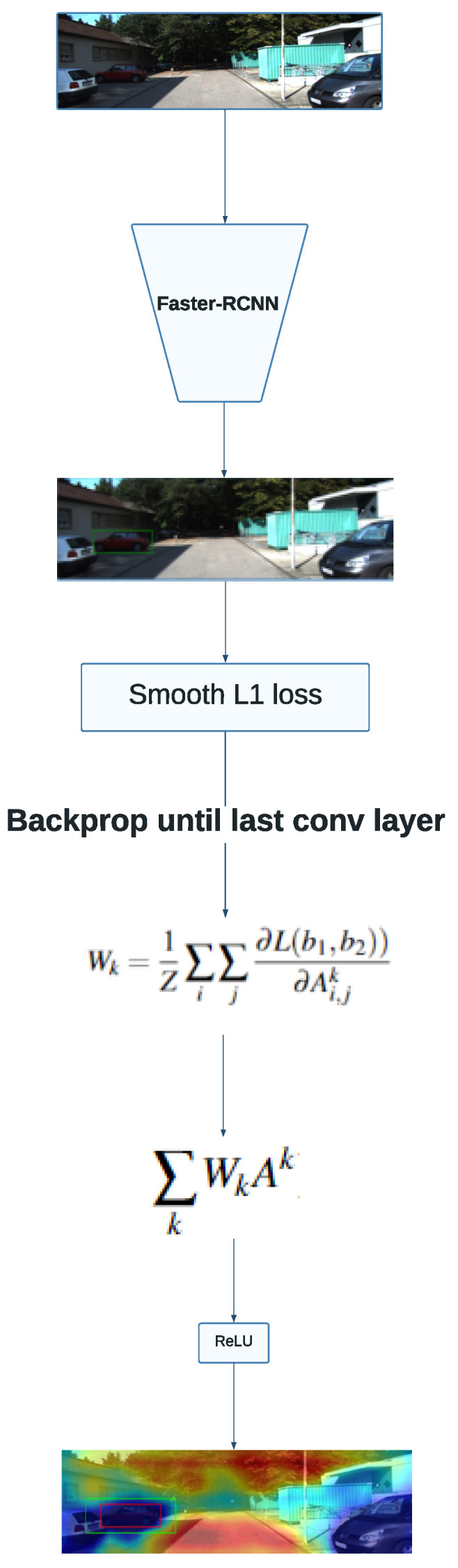
Overview of the Grad-CAM contrastive approach. The red bounding box is the predicted one and the green rectangle is the contrastive example.

**Figure 4 sensors-24-00516-f004:**
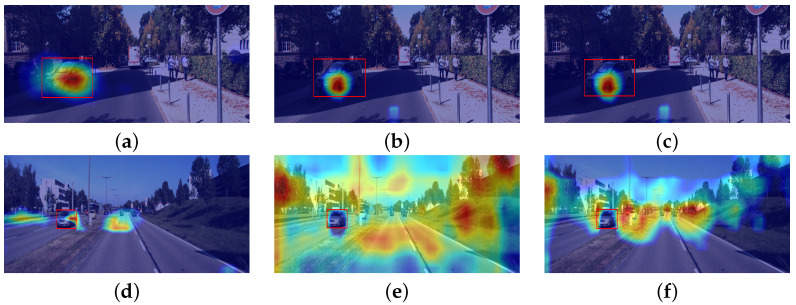
Visual explanations generated with Grad-CAM using scalar transformations in the calculation of the gradients. Images in the third column explain the model that was optimized for localization with a ResNet-101 backbone. Regions in red represent higher activations. (**a**) Distance with RN-50, (**b**) distance with RN-101, (**c**) distance with RN-101 optimized for regression, (**d**) slope with RN-50, (**e**) slope with RN-101, and (**f**) slope with RN-101 optimized for regression.

**Figure 5 sensors-24-00516-f005:**
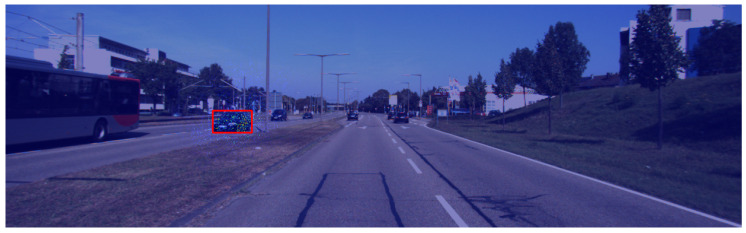
Guided backpropagation example using the ResNet-101 backbone optimized for minimizing localization losses. The gradients were computed with the distance transformation. The green dots close to the red bounding box represent the pixels that are activated in the backpropagation. backpropagation example using the ResNet-101 backbone optimized for minimizing localization losses. The gradients were computed with the distance transformation.

**Figure 6 sensors-24-00516-f006:**
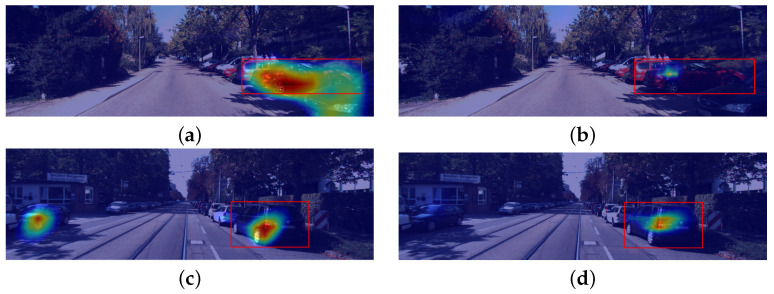
SmoothGrad examples on the Kitti dataset. (**a**,**c**) shows grad-CAM interpretations for the bounding boxes in red. (**b**,**d**) contains Smooth Grad results for the images used for (**a**,**c**), respectively. All heatmaps were generated with the euclidean distance transformation and the best performing model for regression.

**Figure 7 sensors-24-00516-f007:**
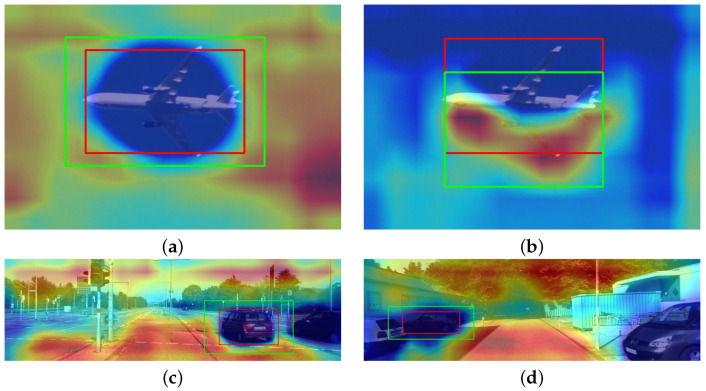
Contrastive explanations with Grad-CAM. Red bounding boxes represent predictions made by the models, while green bounding boxes are contrastive examples. (**a**,**b**) use a model with a ResNet-50 architecture for the Pascal VOC dataset. (**c**,**d**) are explanations in the autonomous driving context using a ResNet-101 backbone.

**Figure 8 sensors-24-00516-f008:**
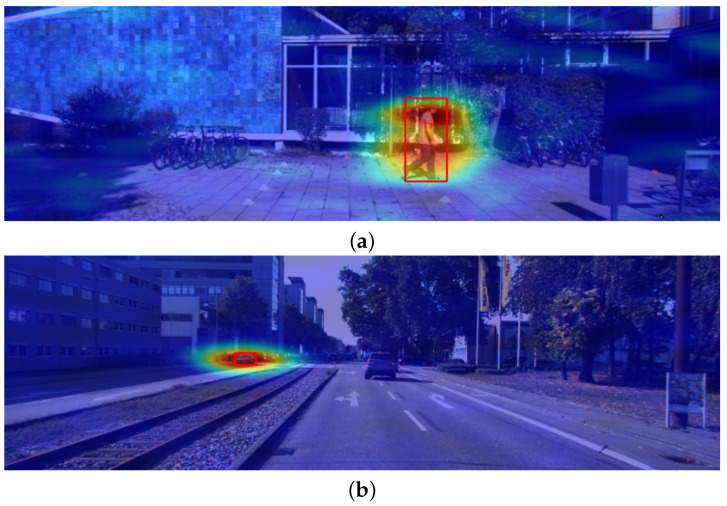
D-RISE explanations. The red rectangles show the predicted bounding boxes. The heatmap highlights in red the most activated regions. (**a**) was generated with localization and classification information. (**b**) solely considered localization for the similarity scores.

**Figure 9 sensors-24-00516-f009:**
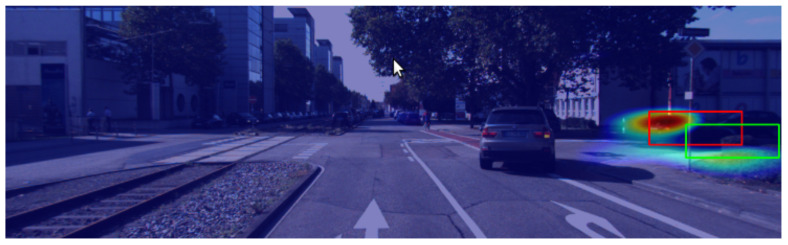
Contrastive explanation using the D-RISE method. The red bounding box represents the predicted bounding box, and the green box shows the contrastive example. The heatmap highlights the regions with higher activations in red.

**Figure 10 sensors-24-00516-f010:**
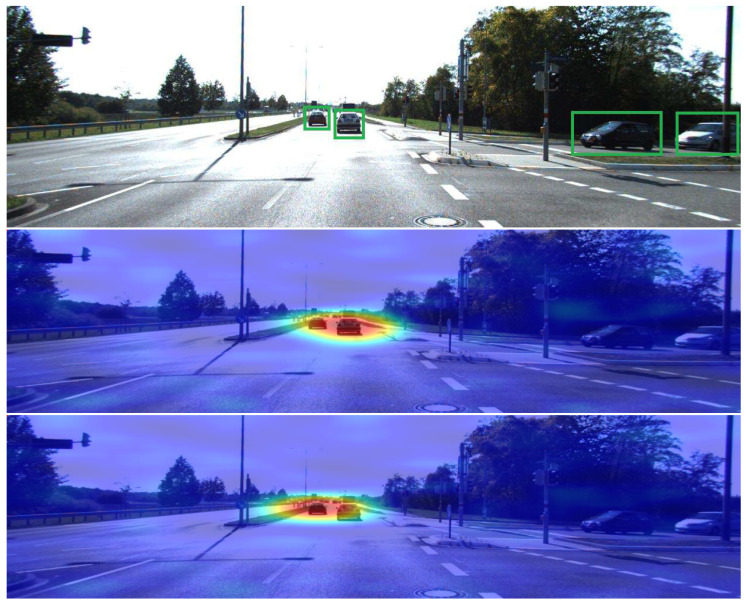
YOLO explanations obtained for two different objects in the same image. The bounding boxes are marked in green. Saliency maps highlight in red the regions with higher activations.

**Table 1 sensors-24-00516-t001:** Exploratory data analysis on datasets.

	Pascal VOC 2012	Kitti
Detections per image (avg)	2.76	6.93
Average object area (pixels)	36,016.50	9096.60
Std object area	45,313.50	17,198.60

**Table 2 sensors-24-00516-t002:** Performance of all models according to mAP and Smooth L1 metrics. Bold values in the table highlight the best model concerning each metric.

Model	mAP_50	Smooth L1
faster-RCNN (ResNet-50)—Kitti	0.72	10.53
faster-RCNN (ResNet-101)—Kitti	**0.74**	10.05
faster-RCNN (ResNet-50)—VOC	0.58	25.72
faster-RCNN (ResNet-101)—VOC	0.60	23.15
faster-RCNN Regression	-	**9.60**

**Table 3 sensors-24-00516-t003:** Pointing Game scores for Grad-CAM on the Kitti dataset. Bold values mean better performance. The ResNet-101* backbone was solely optimized for regression.

PG	Distance Transformation	Slope Transformation
ResNet-50	0.388	0.217
ResNet-101	0.38	0.097
ResNet-101*	**0.439**	0.154

**Table 4 sensors-24-00516-t004:** Pointing Game scores for D-RISE explanations. Bold values mean better performance. The YOLO model (YOLOv5*) contains noisy results, as it evaluated the same number of test images but different samples.

PG Scores	
D-RISE (Localization)	0.75
D-RISE (Localization and Classification)	0.72
D-RISE (Contrastive)	0.21
Grad-CAM	0.44
Guided backpropagation	**0.76**
YOLOv5*	**0.65**

## Data Availability

All datasets used for this study are publicly available. The details regarding the KITTI dataset can be found on [23], and the Pascal VOC can be found on [24]. The results regarding the performance of similar object detectors is also available on the latter references.

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
