# Peer review of "Explaining Bounding Boxes in Deep Object Detectors Using Post Hoc Methods for Autonomous Driving Systems"

_sensors, 2024, doi:10.3390/s24020516_

Round 1

Reviewer 1 Report

Comments and Suggestions for Authors

The paper addresses explainability in object detection frameworks. Current well known approaches like GradCAM, RISE, and SmoothGrad are designed for object recognition where object class is the only variable of interest. In object detection, both class and coordinate bounding boxes are required and must be considered in explainability. The paper addresses an important topic.

I have the following concerns regarding the paper in its current state.

1. Experiments: The paper proposes xAI for object detection. As opposed to classification experiments where there is only one object in the image, there are a number of objects present in detection and algorithms must detect them all. This makes xAI harder. However, in all subjective results, the authors only showcase images where there is only a single object. Both Pascal VOC and Kitti datasets have multiple objects in each scene and the results are far less compelling when showcased on the chosen images. One of the reasons that make the authors work novel is expanding object recognition xAI methods to detection, but this is lost by treating the underlying data as if it were just 1 object. 

 2. Clarity of the method: The paper proposes bounding box regression for backpropagation as its main novelty. This is expanded in lines 163 - 171 for GradCAM. However, this is insufficient and I am unsure where in Eq. 1 both the transformations are present. Are the authors suggesting that T can be either of distance or slope. And importantly, what is the distance or slope against? Presumably, there is no ground truth bounding box, hence what is the distance/slope of the predicted bounding box taken against? The contrastive boxes?

3. Paper structure: The paper expands on xAI methods before jumping into method. However, there is no expansion on object detection. There must be a subsection that expands on object detection and challenges and methods. This might also help to clarify my questions about the method above.

4. Overall readability: There are a number of readability issues throughout the paper. For instance, what is b1 and b2 in Fig 3. 

Some minor concerns include:

1. The conclusion claims that the paper performs an initial study regarding xAI in AVs. However, there are a number of existing papers that, not only conduct initial studies and surveys [1] but also provide more rigorous methods [2]. The authors must conduct a more thorough literature review and describe their novelty to whats out there. 

2. Why the choice of pointing game? What metric of explanations are the authors showcasing in the numerical experiments (importance? relevance? sensitivity? consistency? of explanations). Please see [3-5]

[1] Atakishiyev, Shahin, et al. "Explainable artificial intelligence for autonomous driving: A comprehensive overview and field guide for future research directions." arXiv preprint arXiv:2112.11561 (2021). 

[2] Zemni, Mehdi, et al. "OCTET: Object-aware Counterfactual Explanations." Proceedings of the IEEE/CVF Conference on Computer Vision and Pattern Recognition. 2023.

[3] Murdoch, W. James, et al. "Definitions, methods, and applications in interpretable machine learning." Proceedings of the National Academy of Sciences 116.44 (2019): 22071-22080.

[4] Linardatos, Pantelis, Vasilis Papastefanopoulos, and Sotiris Kotsiantis. "Explainable ai: A review of machine learning interpretability methods." Entropy 23.1 (2020): 18.

[5] AlRegib, Ghassan, and Mohit Prabhushankar. "Explanatory paradigms in neural networks: Towards relevant and contextual explanations." IEEE Signal Processing Magazine 39.4 (2022): 59-72.

Comments on the Quality of English Language

The readability is sometimes hindered because of the writing quality. For instance, the second sentence in the abstract: This field leverages techniques developed for computer vision in other domains 2 for accomplishing perception tasks such as object detection. `This' could refer to AV or deep learning based on previous sentence. 

Reviewer 2 Report

Comments and Suggestions for Authors

In this work, the authors explore explainable AI techniques for the object detection task in the context of autonomous driving.

Comments:

1. The authors state: "This work employs the Faster-RCNN two-stage detector" (page 3, line 129). This is the first time the Faster-RCNN two-stage detector has been mentioned. Since it is the approach used in this paper, the authors should provide more details and references for this architecture. 

2. Same comments for "D-RISE". Mentioned from the abstact and described in few line at page 7. Please provide more details.

3. The authors state: "The difference between the mAP values is related to the number of classes between datasets." (page 4, line 143). Please specify the number of classes for each dataset (in the text or in Table 1).

4. Figure 2 and Figure 3 are clear but it is not clear which is the final approach used in the paper. Please provide a final flowchart or a summary of the proposed method,  as Section 3 is detailed but quite difficult to read.

5. The experimentation focuses on the ResNet architecture. A more complete comparison with the state of the art should be provided through experimentation or discussion on other and/or more recent architectures.

Reviewer 3 Report

Comments and Suggestions for Authors

The author conducted an interpretation of boundary boxes in autonomous driving to solve the black box sub problem of neural networks. But the theme of the article needs to be clearer, and the experimental section should be supplemented appropriately. The specific opinions are as follows:

1. Point by point indicate the contribution of the article in the introduction section.

2. The theoretical part needs to be strengthened.

3. The experimental section can further supplement experiments in different scenarios.

Round 2

Reviewer 1 Report

Comments and Suggestions for Authors

I thank the authors for the changes.

1. I understand the author's argument regarding the limitations of existing visual explanations. This limits the novelty of the proposed work significantly since the changes that the authors have made does not address the explainability in multiple object setting, which is an inherent issue in AVs. The authors state - "In light of this suggestion and some of our previous concerns, we decided to include a more complete explanation by computing individual saliency maps for each object within the same image." Is this limited to section 4.3 only? Maybe a complete D-RISE row, showing the results from this combination (even though its noisy), in Table 3 can help.

2. Even with the rewrites, the novelty of the work itself is not sufficiently represented well in Section 3. At a quick glance, it is hard to see what the authors have done differently to existing methods. For instance, adding lines 210-221 helps in understanding the changes that the authors have made. However, lines 198-206 describes the GradCAM method itself, which is not the author's contributions. A majority of sentences in sections 3.3, 3.4, 3.5 and 3.6 also describe existing work. While this is essential to understanding the author's method, it is misplaced in Section 3. The descriptions of the existing methods (lines 198-206, 229 - 228 etc.) can go in Section 2 as a Background subsection, and Section 3 can concentrate only on the changes that the authors have made to the methods. 

Comments on the Quality of English Language

Section 3 is still unclear. More emphasis on what the authors have done rather than explaining existing methods is required.

Reviewer 2 Report

Comments and Suggestions for Authors

The authors responded to all comments raised, improving the quality of the manuscript.

I have no further comments/suggestions .

Author Response

We thank the reviewer for the comments made regarding the manuscript. Nonetheless, we are still attaching the letter with responses to new concerns raised by the previous version
